# Bird clades with less complex appendicular skeletons tend to have higher species richness

Andrew Brinkworth [1] ✉, Emily Green [2], Yimeng Li[3], Jack Oyston [1,4], Marcello Ruta [2] & Matthew A. Wills[1]

Species richness is strikingly uneven across taxonomic groups at all hierarchical levels, but the reasons for this heterogeneity are poorly understood. It is well established that morphological diversity (disparity) is decoupled from taxonomic diversity, both between clades and across geological time. Morphological complexity has been much less studied, but there is theory linking complexity with differential diversity across groups. Here we devise an index of complexity from the differentiation of the fore and hind limb pairs for a sample of 983 species of extant birds. We test the null hypothesis that this index of morphological complexity is uncorrelated with clade diversity, revealing a significant and negative correlation between the species richness of clades and the mean morphological complexity of those clades. Further, we find that more complex clades tend to occupy a smaller number of dietary and habitat niches, and that this proxy for greater ecological specialisation correlates with lower species richness. Greater morphological complexity in the appendicular skeleton therefore appears to hinder the generation and maintenance of species diversity. This may result from entrenchment into morphologies and ecologies that are less capable of yielding further diversity.

There is a notoriously uneven distribution of species richness across the tree of life, with a disproportionately large number of species concentrated in a relatively small number of higher taxa. Indeed, the same marked asymmetry applies at all taxonomic levels, resulting in markedly imbalanced species distributions across sister groups[1,2]. The reasons for this asymmetry are poorly understood (but see refs. [3,4]). Here, we explore the imbalance of species richness across extant bird groups, and test whether one aspect of their morphological complexity – the morphological differentiation of skeletal elements between fore and hind limbs – correlates with species richness across major avian clades.

We follow McShea[5–7] and McShea & Brandon[8] in conceptualising biological complexity as a function of the number of elements comprising a structure, the grouping and organisation of those elements,

and the relative differentiation between them. A system or structure comprising many parts that are greatly differentiated from one another is deemed more complex than a system or structure with fewer, homogeneous parts in a regular arrangement. While there is no universally agreed definition of complexity[9–12], still less any single metric for its quantification[13], systems with greater complexity typically also require more information to enable their description[14], which our chosen conceptualisation of complexity satisfies. Complexity in this sense is solely a property of physical structure, and not of function. Form and function are inextricably linked, but complexity of form and complexity of function need not be[5]. Thus, we have no a priori expectation that any given ecology or life history should necessitate particular levels of morphological complexity.

[1]Milner Centre for Evolution, Department of Life Sciences, University of Bath, Bath BA2 7AZ, UK. [2]Joseph Banks Laboratories, Department of Life Sciences, University of Lincoln, Green Lane, Lincoln LN6 7DL, UK. [3]Nanjing Institute of Geology and Paleontology, Chinese Academy of Sciences, Nanjing 210008, China. [4]Centre for Integrative Anatomy, Department of Cell and Developmental Biology, University College London, LondonWC1E 6BTUK. ✉e-mail: andrew.r.brinkworth@bath.edu

The appendicular skeleton of birds is morphologically very diverse[15–18]. Birds also show a striking variation in species richness across clades of similar taxonomic rank or with the same most recent common ancestor[19]. For example, nearly 60% of living bird species are Passeriformes, whereas their sister clade – the Psittaciformes – encompass only 3.6% of extant avian diversity[19–21] (Table 1). We note, therefore, that species richness is not necessarily expected to correlate with clade age[22]. The origin of birds from theropods, and the remarkable specializations in the appendicular skeleton that accompanied their subsequent radiation[16,17,23], offer an excellent system for analysing the macroevolutionary relationships between diversity and complexity.

Morphological variety (disparity) is often a poor proxy for taxonomic diversity, and the decoupling of species richness (diversity) from the morphological disparity of species within clades or time bins is well established[24]. Most comparisons of diversity and disparity in macroevolutionary contexts have been implemented using palaeobiological data and feature extinct groups[24–29], but there are also striking examples of decoupling in modern organisms. For example, extant afrotherian mammals have a modest diversity compared with many other mammalian clades, but also relatively high disparity: from elephants to elephant shrews, and from manatees to aardvarks. This disparity is reflected, for example, in terms of the wide variety of their vertebral formulae[30]. Clade disparity is therefore unlikely to offer much explanatory power for the heterogeneity of species numbers across clades, or vice versa.

Different modes of selection may have opposite effects upon species diversity within clades. Negative frequency-dependent selection[31,32] tends to promote the evolution of greater diversity[33], while stabilising selection tends to have a homogenising effect (e.g., ref. 34), thereby inhibiting diversification. Some theoretical models[35,36] posit that more complex phenotypes have a greater probability of maintaining phenotypic diversity. Phenotypes with higher dimensionality (i.e., with more variable components) are considered to have

**Table 1 | Summarised data for each ordinal-level clade used in analyses**

| Clade | Species Richness | Mean Complexity (± σ) | Sample size (n) | Foraging niches | Trophic niches | Habitat types |
|---|---|---|---|---|---|---|
| Accipitriformes | 252 | 0.502 (0.250) | 43 | 8 | 5 | 8 |
| Anseriformes | 178 | 0.529 (0.112) | 132 | 8 | 4 | 7 |
| Apodiformes | 465 | 0.965 (0.186) | 14 | 3 | 2 | 5 |
| Bucerotiformes | 72 | 0.563 (0.128) | 4 | 2 | 2 | 3 |
| Caprimulgiformes | 132 | 1.012 (0.487) | 7 | 4 | 2 | 4 |
| Cariamiformes | 2 | 1.077 (0.157) | 2 | 1 | 1 | 2 |
| Cathartiformes | 7 | 0.726 (0.184) | 6 | 1 | 1 | 4 |
| Charadriiformes | 379 | 0.624 (0.355) | 131 | 6 | 3 | 7 |
| Ciconiiformes | 19 | 0.502 (0.159) | 11 | 1 | 2 | 4 |
| Columbiformes | 348 | 0.378 (0.106) | 57 | 3 | 3 | 5 |
| Coraciiformes | 183 | 0.944 (0.262) | 13 | 5 | 3 | 6 |
| Cuculiformes | 147 | 0.454 (0.392) | 4 | 2 | 1 | 3 |
| Eurypygiformes | 2 | 0.547 (0.530) | 2 | 2 | 2 | 2 |
| Falconiformes | 66 | 0.385 (0.133) | 27 | 8 | 3 | 6 |
| Galbuliformes | 54 | 0.942 (0.137) | 3 | 2 | 1 | 2 |
| Galliformes | 295 | 0.562 (0.157) | 63 | 6 | 5 | 6 |
| Gaviiformes | 5 | 0.794 (0.064) | 5 | 1 | 1 | 1 |
| Gruiformes | 192 | 0.586 (0.217) | 22 | 2 | 3 | 3 |
| Opisthocomiformes | 1 | 0.321 (-) | 1 | 1 | 1 | 1 |
| Otidiformes | 26 | 0.363 (0.062) | 3 | - | 1 | 2 |
| Passeriformes | 6386 | 0.387 (0.215) | 127 | 12 | 5 | 8 |
| Pelecaniformes | 114 | 0.400 (0.354) | 42 | 3 | 2 | 4 |
| Phaethontiformes | 3 | 1.396 (0.121) | 3 | 1 | 1 | 1 |
| Phoenicopteriformes | 6 | 0.918 (0.120) | 3 | 2 | 2 | 2 |
| Piciformes | 378 | 0.214 (0.138) | 8 | 4 | 2 | 3 |
| Podicipediformes | 22 | 0.507 (0.081) | 12 | 2 | 1 | 1 |
| Procellariiformes | 138 | 0.949 (0.240) | 69 | 5 | 2 | 1 |
| Psittaciformes | 387 | 0.659 (0.124) | 89 | 4 | 4 | 4 |
| Pterocliformes | 16 | 0.495 (0.066) | 3 | 1 | 1 | 2 |
| Sphenisciformes | 18 | 1.032 (0.076) | 13 | 1 | 1 | 1 |
| Strigiformes | 234 | 0.452 (0.154) | 14 | 4 | 2 | 5 |
| Struthioniformes | 13 | 1.863 (0.765) | 7 | 2 | 3 | 3 |
| Suliformes | 59 | 1.175 (0.479) | 33 | 3 | 1 | 3 |
| Tinamiformes | 46 | 0.391 (0.095) | 7 | - | 1 | 3 |
| Trogoniformes | 43 | 0.703 (0.060) | 3 | 2 | 3 | 2 |

Information includes an estimate of clade species richness, mean complexity score (with standard deviation [σ]), sample size, and the number of foraging niches, trophic niches, and habitat types that their sampled members occupy. Species occupying the "Omnivore" trophic niche are not classified for a foraging niche, resulting in clades with omnivorous species occupying more trophic than foraging niches in some cases (Struthioniformes, Trogoniformes, Gruiformes).

greater complexity. If the interactions between those dimensions have an impact on fitness, the strength of negative frequency-dependent selection necessary to overcome stabilising selection is greatly reduced. This model has been extended to account for underlying ecological resource competition[36], and supports the notion that adaptive diversification is more probable in complex phenotypes.

Theoretical work by Orr[37] predicts that more complex organisms evolve and adapt more slowly than less complex organisms. However, whether this implies that groups of organisms with higher mean complexity diversify more slowly depends upon whether bursts of speciation and concurrent net diversification are adaptive or non-adaptive[38]. Whilst much diversification in birds is believed to be non-adaptive[39], there are also some conspicuous adaptive radiations, including Darwin's finches, Hawaiian honeycreepers[40] and Madagascan vangas[41]. One possibility is that adaptive radiations occur more frequently in clades of lower complexity, owing to faster rates of adaptive evolution, and that these clades have greater species richness as a result. However, net diversification is as much a function of rates of extinction as of speciation. The link between Orr's prediction and extinction rates could result from variable extinction risks, as more slowly adapting species are at greater risk in more rapidly changing environments[42,43]. Moreover, narrow habitat niches are associated with a greater extinction risk[44], which is also consistent with specialists being at greater risk of diversity loss.

Differences in morphological complexity may, therefore, provide some explanation for uneven species richness, as greater complexity may promote more rapid diversification. Alternatively, greater morphological complexity may act as a hindrance to adaptive radiation and confer a greater risk of extinction, thereby acting as an inhibitor of rapid diversification. This leaves us with no clear consensus on what to expect from the empirical relationship between complexity and diversity. Here, we quantify an aspect of complexity in the avian appendicular skeleton, and address the null hypothesis that limb skeleton complexity is not correlated with the species richness of clades across living birds. Our investigation tests these hypotheses, utilizing the taxonomically, morphologically, and ecologically diverse radiation of these well-studied and iconic vertebrates.

In our model system, the fundamental axes of phenotypic variation are the lengths of the six major elements of the appendicular skeleton (excluding the phalanges): namely the femur, tibiotarsus, tarsometatarsus (hindlimb), humerus, ulna, and carpometacarpus (forelimb). Further axes of variation include the relative difference in length between limb pairs, and the relative proportions of equivalent elements (i.e., stylopod, zeugopod, and autopod) between limb pairs. Considered together, these variables allow us to index the "differentiation" component of morphological complexity (*sensu* refs. [5–8]), with the number of constituent parts remaining constant. Whilst the number of elements is constant in our system, this need not always be the case. Different groups of arthropods, for example, can have highly divergent numbers of limb pairs. Differentiation here is analogous to the concept of morphological disparity. The key distinction is that in this case we refer to the morphological variation within a single individual, not across groups of species, as is the case with disparity.

A lack of complexity given our chosen definition is, therefore, exemplified by homogeneity. The lowest complexity scores (see Methods) will be achieved when both limb pairs are of the same length, with corresponding elements (stylopods, zeugopods, and autopods) between the two limbs being the same length. Importantly, this does not necessitate that elements within a limb are identical in length, merely that corresponding elements across the two limb pairs are identical. Hence, there are an unlimited number of ways of obtaining the minimum index of complexity, with stylopods, zeugopods and autopods of very different lengths and proportions, provided these are the same in both limb pairs.

The appendicular skeleton is an excellent study system for two reasons. Firstly, variability in limb skeleton morphology is broad and well documented[15–17]. Extant birds exhibit a wide range of proportional and size differences between limb pairs[18], as do their extinct relatives[45]. Secondly, within- and between-limb differences in element proportions are frequently associated with ecological specialisation, and are underpinned by well understood development mechanisms[46]. One of the stipulations of some theoretical models[35,36] is that axes of morphological variation must interact to influence the fitness of the whole organism in order to promote diversification. It is well established that limbs with different ecological or biomechanical functions often exhibit differences in the proportions of equivalent elements between limb pairs[18,47,48], reflecting energetic trade-offs in development[49]. Further, functionally related elements of the avian limb skeleton show a signal of evolutionary integration[50]. This evolutionary, ecological, and ontogenetic intertwining of limb morphology is evidence that interactions between our axes of variation have an impact on fitness[35,36], and that our metric of complexity, indexed as differentiation between the fore and hind limbs, has biological relevance.

Here, we show that the mean morphological complexity of the limb skeleton in bird clades correlates significantly and negatively with their species richness. Further, we find significant relationships between the foraging, trophic, and habitat niches occupied by species, and their complexity. Finally, we find that clades with greater mean complexity tend to occupy fewer ecological foraging niches, as well as being less species rich. These results suggest that greater morphological complexity acts as a hindrance to taxonomic diversification, potentially via greater ecological specialisation associated with those more complex phenotypes. Our results are also congruent with theoretical models suggesting that complexity will slow rates of adaptive evolution[37], and are contrary to models suggesting complexity should enhance diversification[35,36].

## Results

### Allometric variation can impact inferred complexity scores

Our index of complexity for a given species was derived as the Euclidean distance between fore and hind limbs in a three-dimensional morphospace, with one axis representing the lengths of each limb segment (stylopod, zeugopod, autopod), and each species therefore represented by two points (forelimb and hindlimb). Given the influence of body size upon many aspects of organismal biology, the removal of body-size related variation in bone lengths prior to further analysis was critical. We achieved this in two steps, one each to remove the effects of isometric and allometric variation. We first removed isometric size variation, that would otherwise inflate scores among larger-bodied species (see Methods; Supplementary Fig. 1), by expressing the length of each element as a proportion of the mean length for that species. We then applied a phylogenetic regression to these isometrically-transformed relative mean bone lengths, against body mass estimates for each species. From this we obtained phylogenetic residuals for each relative element length, independent of species body mass. This regression has the effect of removing allometric variation in relative bone lengths. Residual relative lengths of appendicular skeletal elements are highly variable across species, and between limb pairs (Fig. 1a–c). Data were compiled for a sample of 983 extant species (Supplementary Data 1).

Complexity scores derived in this way were found to exhibit no significant correlation with body mass in a phylogenetic generalised least-squares analysis (PGLS, $p = 0.3024$, $F = 1.065$ on 1 & 981 d.f., adjusted $R^2 = 6.588 \times 10^{-5}$, $N = 983$; Fig. 2a), confirming the successful removal of the influence of body size on inferred complexity scores. Complexity scores derived directly from the isometrically-transformed values representing relative bone lengths, without the regression step to correct for allometry, were found to correlate

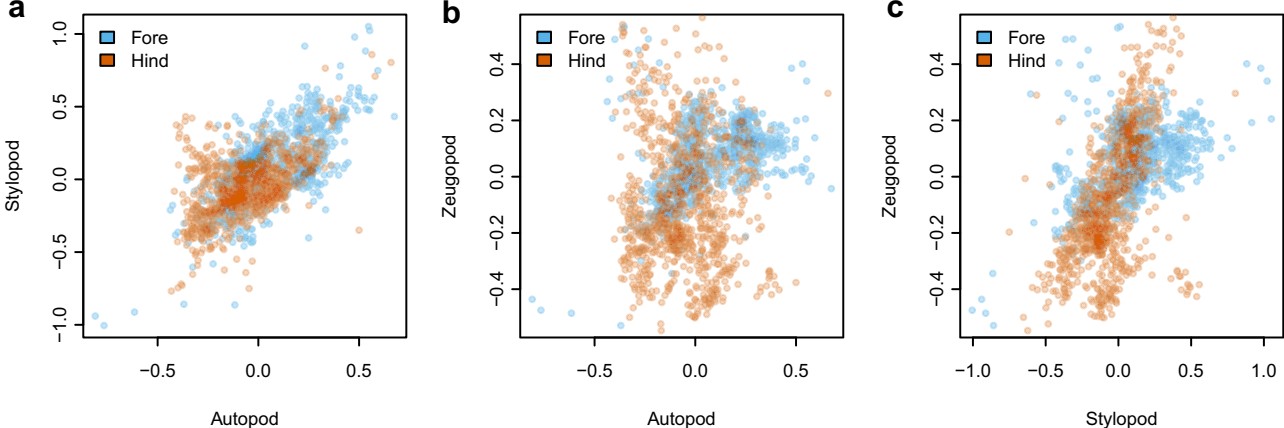

**Fig. 1 | Bivariate relationships between the relative differentiation of limb segments in pairwise comparisons.** Values indicated are phylogenetic residuals from a PGLS model of isometrically-transformed mean bone lengths against estimates of species body masses. In all cases, the forelimb is represented in blue, and the hindlimb in orange. **a** The autopod (tibiotarsus and ulna) and stylopod (femur and humerus), **b** the autopod and zeugopod (tarsometatarsus and carpometacarpus), **c** the zeugopod and stylopod. "Hind" = hindlimb, "Fore" = forelimb.

significantly with body mass (PGLS, $p = 4.763 \times 10^{-6}$, F = 21.17 on 1 & 981 d.f., adjusted $R^2 = 0.0201$, $N = 983$; Fig. 2b).

To avoid conflating patterns of body size evolution with patterns of evolution in complexity, we use and refer to the complexity scores calculated from residual relative mean bone lengths (Fig. 2a) in all subsequent analyses.

## Mean complexity and species richness are negatively correlated across clades

Mean complexity is highly variable across extant avian orders (Table 1; Fig. 3). The greatest mean complexity is inferred for the composite "Struthioniformes" clade (ostriches, cassowaries, rheas, emus, and kiwis), and the lowest for the Piciformes (woodpeckers and allies). We found a significant negative correlation between the mean $\log_{10}$-transformed complexity score of orders and their $\log_{10}$-transformed species richness (PGLS, $p = 0.0134$, F = 6.86 on 1 & 32 d.f., adj.$R^2 = 0.1508$, $N = 34$), such that more complex orders tend to be less species rich (Fig. 4). A strong phylogenetic signal is inferred for complexity at the ordinal level (K = 2.259), indicating that more closely related clades are more similar to each other than expected under a purely Brownian model (for which K = 1,[51]). The goodness-of-fit of the PGLS model, as approximated by the adjusted $R^2$, is relatively small. Hence, despite the significant effect of complexity, other unmodelled parameters must also influence the species richness of clades.

Passeriformes are a marked outlier in terms of their exceptionally high species richness (Fig. 4). We therefore refitted this model with Passeriformes excluded, to ensure they were not unduly influencing our results. We found a similar significant negative correlation between mean clade complexity and species as with the full data set (PGLS, $p = 0.0242$, F = 5.621 on 1 & 31 d.f., adj.$R^2 = 0.1262$, $N = 33$).

## Complexity is significantly, but weakly, correlated with trophic niche, foraging niche, and habitat type

Firstly, we found a significant relationship between the complexity of species and their ecological foraging niche (PGLS, $p = 3.359 \times 10^{-4}$, F = 2.192 on 29 & 743 d.f., adj.$R^2 = 0.0429$, $N = 773$). However, the explanatory power of this PGLS model (approximated by the adjusted $R^2$) was very low. Species occupying aquatic plunge diving (e.g., *Sula*, boobies) and aerial aquatic (e.g., *Fregata*, frigatebirds) niches are associated with the greatest mean complexity scores, whereas the lowest scores were observed in perching vertivores (e.g., *Harpia harpyja*, harpy eagle) and aerial vertivores (e.g., *Falco peregrinus*, peregrine falcon). Mean values for each foraging niche are provided in

Supplementary Table 2. Complexity has a relatively weak phylogenetic signal at the species-level (K = 0.698), in contrast with the very strong signal inferred at the higher taxonomic level.

Secondly, there was a significant correlation between complexity and trophic niche (PGLS, $p = 6.504 \times 10^{-5}$, F = 3.928 on 9 & 860 d.f., adj.$R^2 = 0.0294$, $N = 870$; Supplementary Table 3). The trophic niche data are coarser than foraging niche data, containing fewer categories. Further, some trophic niches, as defined, contain a greater diversity of sub-niches. For example, the "Aquatic predator" trophic niche is sub-divided into 6 foraging niches, whereas the "Generalist" and "Omnivore" trophic niches contain no further sub-division of foraging niches.

Thirdly, we found a significant correlation between complexity scores and habitat type (PGLS, $p = 0.0213$, F = 2.182 on 9 & 858 d.f., adj.$R^2 = 0.0121$, $N = 868$). This model has a low explanatory power, which could reflect its coarseness and modest number of categories ($N = 10$). The greatest complexity scores are observed for marine species, and the lowest for grassland species (Supplementary Table 4).

We further found a significant, but very weak, positive correlation between complexity scores and Kipp's distance across species (PGLS, $p = 4.275 \times 10^{-5}$, F = 16.92 on 1 & 859 d.f., adj.$R^2 = 0.0182$, $N = 861$; Supplementary Fig. 2; Supplementary Table 6). Kipp's distance has links to aerodynamics and flight ability[52], and this result suggests that more complex species may tend to be better flyers. We note, however, that the greatest complexity scores are associated with both the greatest and lowest Kipp's distances. We inferred from diagnostic plots that the fit of this model was unreliable, despite the removal of several outliers. We do not consider this result to be a key finding, but report it here given its relevance to the discussion.

## Members of more complex clades tend to occupy fewer foraging niches and habitats

We use the number of ecological niche categories occupied by the sampled members of a clade as an inverse proxy for the degree of ecological specialisation within that clade. Occupation of fewer niches indicates a greater degree of specialisation, and vice versa. Passerines occupy the largest number of niches, whilst several clades occupy just a single foraging niche (e.g., Sphenisciformes [penguins], Cathartiformes [New World vultures], Ciconiiformes [storks and ibis]; Fig. 5a; Table 1). Further, we found a significant negative correlation between the number of niches occupied, and mean ordinal complexity (PGLS, $p = 5.779 \times 10^{-3}$, F = 8.834 on 1 & 30 d.f., adj.$R^2 = 0.2017$, $N = 32$; Fig. 5a). This demonstrates a tendency for clades with a greater mean

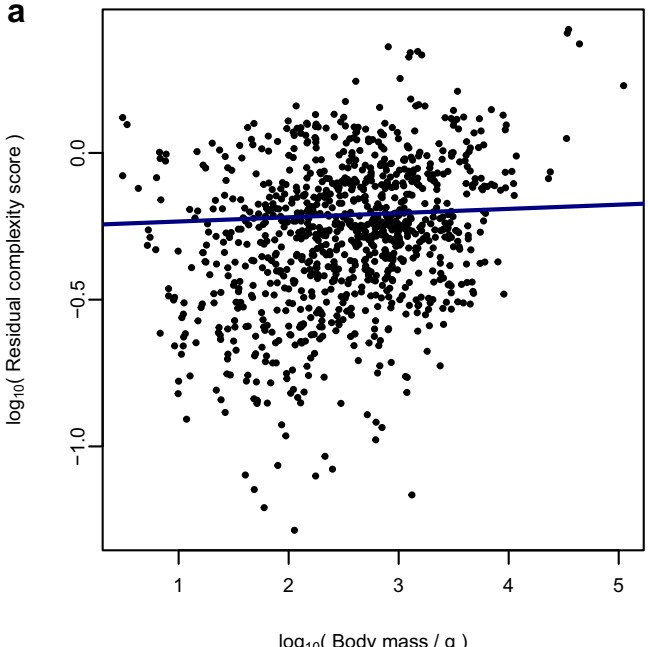

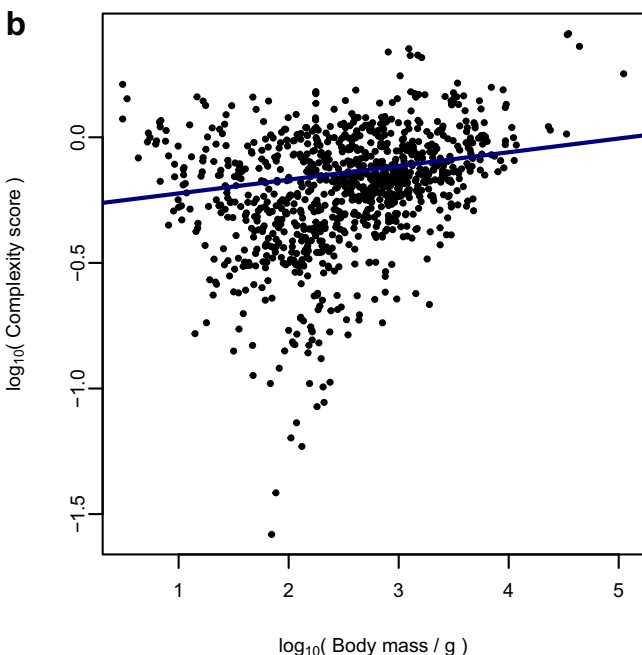

**Fig. 2 | Correlations between species body mass and complexity scores.** Plots showing correlations between **a**) complexity scores calculated with an additional step to remove allometric variation and differences linked to body mass, and **b**) complexity scores derived from isometrically-transformed mean bone lengths, without proper control for allometric variation. Regression lines are those obtained from PGLS analyses of the data. The scores shown in panel A are labelled as "residual" complexity scores in this instance, to denote the difference in data handling. Complexity scores and body masses were $log_{10}$-transformed prior to model-fitting. The correlation in **a**), while positive, is non-significant, whereas the one in **b**) is both positive and significant. This indicates that the removal of allometric variation in mean bone lengths is critical in fully removing the influence of trends in body mass from these complexity data.

limb complexity to occupy fewer foraging niches, and therefore to have greater ecological specialisation. This pattern was not retrieved when we used the coarser trophic niche data (PGLS, $p = 0.3518$, F = 0.893 on 1 & 32 d.f., adj.$R^2 = -3.254 \times 10^{-3}$, $N = 34$; Fig. 5b).

We found a marginally significant negative correlation between the number of different habitat types occupied by members of a clade and the mean complexity of those members (PGLS, $p = 0.0645$, F = 3.668 on 1 & 32 d.f., adj.$R^2 = 0.0748$, $N = 34$; Fig. 5e). Passerines and Accipitriformes (hawks and allies) jointly occupied the greatest number of habitat types ($N = 8$). Clades occupying only a single habitat type were all "marine" (Phaethontiformes [tropicbirds], Procellariiformes [petrels, albatrosses, and allies], Gaviiformes [gavies], Podicipediformes [grebes], and Sphenisciformes [penguins]).

### Clades occupying more trophic niches and habitat types tend to be more species-rich than expected by chance

Orders with a greater species richness tend to occupy a greater number of foraging niches (PGLS, $p = 9.680 \times 10^{-7}$, F = 37.59 on 1 & 30 d.f., adj.$R^2 = 0.5413$, $N = 32$; Fig. 5c). The mega-diverse Passeriformes occupy the greatest number of foraging niches ($N = 12$). By contrast, the Phaethontiformes (tropicbirds, 3 species, "aquatic plunge"), Cariamiformes (seriemas, 2 species, "invertivore ground"), Gaviiformes (loons and allies, 5 species, "aquatic dive"), and Cathartiformes (New World vultures, 7 species, "scavenger ground") all occupy single niches. Some of these clades also exhibit a relatively high mean complexity (Fig. 3).

It might be expected that more diverse clades would encompass a greater diversity of dietary ecologies purely by virtue of their larger sample sizes. To estimate the nature of sample size effects in our analyses, we used a randomisation approach in which tip data were randomly reallocated into groups of equal size and number to the original clades. We found that significant positive correlations between species richness and foraging niche diversity are the null expectation from randomly distributed data (Supplementary Fig. 3A). We then compiled the coefficients from the models for randomly assigned data, and compared these to the coefficients of the empirical models. For foraging niche data, we found that the empirical model had a significantly greater y-intercept (randomisation test, $p < 0.001$; Supplementary Fig. 3B) and significantly lower slope than expected at random (randomisation test, $p < 0.001$; Supplementary Fig. 3C), indicating the inflation of niche diversity in clades of greater species richness is not as pronounced in the real data as would be expected by chance. The empirical model does not, however, explain any more variance than the randomised ones (randomisation test, $p = 0.756$; Supplementary Fig. 3D). Note that these are not exact $p$-values, as randomisation tests do not provide an exact $p$-value unless all possible permutations of the data are assessed (see Methods).

The same tendency was found when we used the trophic niche data, namely a significant positive correlation between clade species richness and the number of trophic niches that the sampled members of those clades occupy (PGLS, $p = 1.149 \times 10^{-4}$, F = 19.30 on 1 & 32 d.f., adj.$R^2 = 0.3567$, $N = 34$; Fig. 5d). The weaker explanatory power of this model compared to that of the foraging niche model may be a function of coarser resolution. As for the foraging niche data, significant positive correlations between species richness and trophic niche diversity were found to be the null expectation (Supplementary Fig. 4A). The empirical model was found to imply a significantly lower y-intercept (randomisation test, $p < 0.001$; Supplementary Fig. 4B), and a significantly greater slope than expected at random (randomisation test, $p < 0.001$; Supplementary Fig. 4C), and explain a significantly lower proportion of variance than the randomised models (randomisation test, $p = 0.001$; Supplementary Fig. 4D). This indicates that more diverse clades tend to occupy more trophic niches than expected by chance, and vice versa for depauperate clades. We might infer from this that species-poor clades are particularly conserved in their trophic ecologies.

The number of habitat types occupied by members of a clade correlates positively and significantly with their species-richness, such that more species rich clades tend to occupy more habitat types (PGLS,

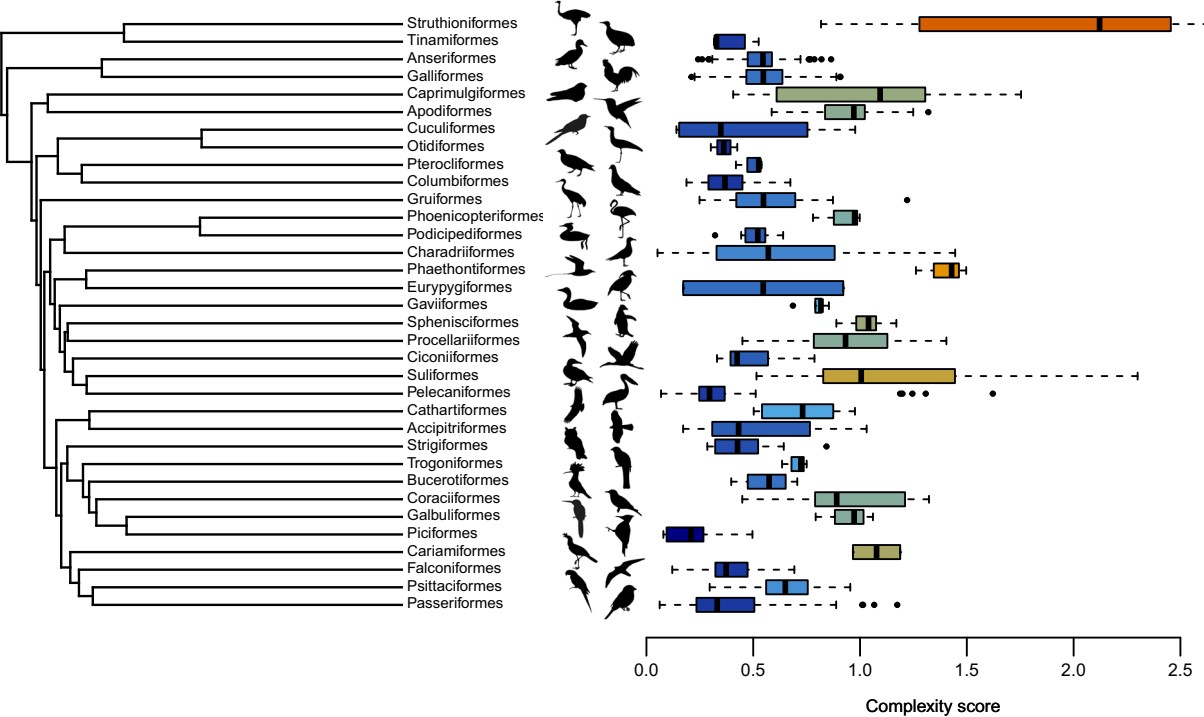

**Fig. 3 | Complexity scores for each ordinal-level clade used in analyses.** Boxplots of complexity scores for each clade are shown in alignment with the ordinal-level tree used in phylogenetically informed statistical analyses. Boxplots indicate the median, upper and lower quartiles, and upper and lower extremes of complexity sores for species in each clade. Outliers are indicated. Taxa are indicated by name, and silhouettes are matched to their corresponding tips. Variation in mean complexity scores among clades are also indicated by colour of the boxplots, with blues indicating lower scores and oranges higher scores. Data are shown for 34 ordinal-level taxa, containing between 2 and 132 sampled species. A full summary of N values for each group are provided in Table 1. The highly variable nature of limb complexity across extant Aves is apparent, with the Struthioniformes having the highest inferred mean score, and the Piciformes the lowest. Silhouettes are from phylopic.org. Under a CC0 1.0 Universal Public Domain Dedication: *Accipiter nisus,*

*Archilochus colubris, Coragyps atratus, Falco peregrinus*, and *Passer domesticus* by Andy Wilson; *Anodorhynchus hyancinthinus* by Zitan Song; *Asio, Caprimulgus, Columba, Coracias garrulus,* Cuculidae, *Dryocopus, Larus, Pelecanus, Phoenicopterus roseus, Podiceps cristatus, Rhynochetos jubatus, Trogon*, and *Upupa epops* by Ferran Sayol; *Eudyptes chrysolophus* by Alexandre Vong; Galbulidae by Frederico Degrange; *Gallus gallus* by Steven Traver; *Grus canadensis* by Lauren Anderson; *Mycteria americana* by Mathieu Basille; *Phaethon lepturus* by Marie-Aimee Allard; *Puffinus griseus* by Juan Carlos Jeri; *Sula granti* by Beth Reinke. Under a Public Domain Mark 1.0 License: *Cariama cristata* and *Tinamus major* by George Edward Lodge (vectorised by T. Micheal Keesey; *Gavia immer* by Marie Attard; *Pterocles guttaralis* by T. Micheal Keesey. Under an Attribution-ShareAlike 3.0 Unported License: *Ardeotis nigriceps* by L.Shyamal (vectorised by T. Micheal Keesey); *Struthio camelus* by Martin Martyniuk (vectorised by T. Micheal Keesey).

$p = 3.994 \times 10^{-5}$, F = 22.65 on 1 & 32 d.f., adj.$R^2$ = 0.3962, $N$ = 32; Fig. 5f). Many of the clades that occupy just a single habitat type are also relatively species poor, especially when compared to habitat and species diverse clades like the Accipitriformes, Anseriformes (water-fowl), Charadriiformes (shorebirds), and particularly the passerines. When considering habitat type data in a randomisation approach, significant positive correlations were again found to be the expectation (Supplementary Fig. 5A). We found that the empirical model implied significantly greater y-intercept ($p < 0.001$; Supplementary Fig. 5B) and slope ($p < 0.001$; Supplementary Fig. 5C) coefficients than were expected at random, but explained a lower proportion of variance than the randomised models ($p = 0.001$; Supplementary Fig. 5D). This implies that clades occupy a greater diversity of habitat types than we would expect by chance.

**Mean complexity and ecological specialisation of clades simultaneously contribute to their species richness**
Using a multiple PGLS model, we found that an additive combination of mean clade complexity and the number of occupied habitat types was a strongly significant predictor of clade species richness (multiple PGLS, $p = 2.192 \times 10^{-5}$, F = 15.47 on 2 & 31 d.f., adj.$R^2$ = 0.4672, $N$ = 34). The number of occupied habitat types was inferred to have a strongly significant positive effect (slope = 1.6049, $p = 1.568 \times 10^{-4}$), and mean complexity a significant negative effect (slope = −0.9884, $p = 0.0249$).

When the number of trophic niches occupied by members of clades was used in this model in place of the number of habitat types, we again found a highly significant result (multiple PGLS, $p = 3.365 \times 10^{-5}$, F = 14.62 on 2 & 29 d.f., adj.$R^2$ = 0.4523, $N$ = 34). In this instance, the inferred effect of the number of occupied trophic niches was strongly significant and positive (slope = 0.3363, $p = 2.213 \times 10^{-4}$), and the effect of mean complexity was strongly significant and negative (slope = −1.2482, $p = 5.616 \times 10^{-3}$).

Finally, when the number of foraging niches occupied by members of each clade was used as a proxy for their ecological plasticity, we again found a strongly significant correlation with species richness (multiple PGLS, $p = 1.499 \times 10^{-6}$, F = 22.06 on 2 & 29 d.f., adj.$R^2$ = 0.5761, $N$ = 32). In this combination, we found that the inferred effect of the number of occupied foraging niche categories had a strongly significant positive effect on species richness (slope = 0.1974, $p = 4.604 \times 10^{-6}$), while mean complexity within clades had a marginally significant negative effect (slope = −0.8899, $p = 0.0734$).

We do not assert that occupying a greater number of ecological niches is a direct driver of greater diversity. The purpose of including this among the explanatory variables was to establish that the sign and significance of the correlations of complexity and ecological specialisations with species richness are maintained in combination, as they are in pairwise comparison.

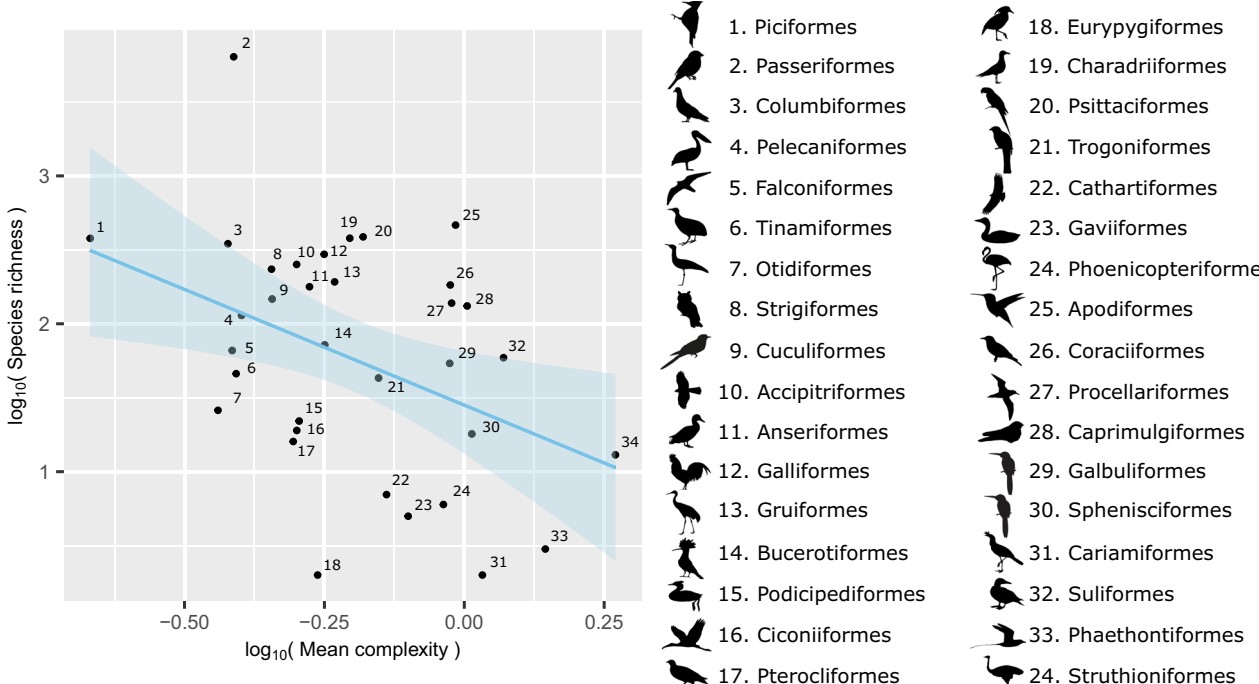

**Fig. 4 | Correlation between mean clade complexity and species richness.** There is a significant negative correlation between mean clade complexity scores and species richness. Standard error is shown as blue shading around the regression line. Both variables were log₁₀-transformed prior to analysis, to improve model reliability. Despite the wide scatter, an inverse relationship is visible. Clades are indicated by number, with corresponding clade names and representative silhouettes shown in the right-hand side of the panel. Silhouettes are from phylopic.org, used under the same licenses as stated in the legend for Fig. 3.

## Discussion

We found evidence that lower clade diversity correlates with greater complexity of the appendicular skeleton across the phylogeny of extant birds and that greater complexity correlates with a tendency to occupy fewer dietary and habitat niches. These findings seem most intuitively to align with the predictions of Orr's[37] models, which demonstrated that more complex species should be slower to evolve and adapt. In contrast, and despite theoretical work predicting it[35,36], we find no evidence of a positive correlation between clade diversity and the morphological complexity of its constituent species when using indices derived from the limb differentiation of birds. There are several possible explanations for these findings.

Bell et al.[18] suggested that a strong phylogenetic signal in the ratios of limb bone lengths may result from lineage-specific developmental trajectories and constraints, while Watanabe[53] has demonstrated these differences do exist amongst groups of waterbirds. Constrained trajectories may limit the realised diversity of complexity of limb skeleton configurations, independent of other factors. However, while a strong phylogenetic signal in relative limb bone proportions at a higher taxonomic level suggests the possible occurrence of such constraints, it is not conclusive evidence. A strong phylogenetic signal can arise from clade-specific developmental differences, but this also arises from other factors.

The structure of the limb skeleton, and by extension its complexity, has obvious ecological and functional implications across avian diversity, and throughout individual life histories. For example, the developmental timing of flight acquisition and the associated parenting behaviours[49], as well as adult flying style[54] are all linked to variable morphology of the limb skeleton. We found a significant relationship between limb skeleton complexity and ecological foraging niches (as defined in ref. 55), particularly in aquatic species. Many ocean-going taxa, such as tropicbirds (Phaethontiformes), and frigatebirds (Suliformes: Fregatidae; Fig. 3) have conspicuously longer wings and shorter legs than the majority of other species. Our results concur with previous work that found the morphology of the appendicular skeleton to be strongly linked to ecology and locomotion[18,46,47]. The low goodness-of-fit (indicated by a low R²) is consistent with comparisons of morphology and ecology in other anatomical regions[56].

It is difficult to define and measure ecological specialisation[57], and here we use the number of niches occupied by the sampled members of a clade as a simple proxy (the fewer niches occupied by members of a clade, the more specialised we consider it to be). Orders with more complex limb skeleton morphologies tended to occupy fewer foraging niches (Fig. 5a) and habitat types (Fig. 5e), and orders occupying fewer niches and fewer habitats tended to be less species-rich (Fig. 5b, Fig. 5f). Also of note, we found that clades tend to occupy fewer foraging niches than would be expected based on randomisation of the data (Supplementary Fig. 3). These relationships were not found when we used the more coarsely resolved trophic niche data (Fig. 5c, d; Supplementary Fig. 4). Therefore, greater complexity in the appendicular skeleton correlates with greater ecological specialisation, and this specialisation also correlates with lower diversity. This suggests that more complex configurations of the appendicular skeleton engender entrenchment within a particular foraging niche, or small set of similar niches.

There are well-known trade-offs between fore- and hindlimb morphology for different locomotory strategies[49]. Such trade-offs are also important in certain ecogeographic contexts, such as island colonisation[58]. For example, many species that are adapted for prolonged flight have reduced legs[48], whereas flightless species often have reduced wings[58-60]. Such specialisations can influence dispersal ability[61], and inhibit further adaptation or diversification in the face of changing environments. The reduced wings of the large-bodied ratites result in flightlessness and, therefore, a lesser ability to disperse or invade non-terrestrial foraging niches. The short legs of frigatebirds would leave them particularly vulnerable to predators on the ground[49]. While dispersal ability is difficult to suitably proxy in analyses of

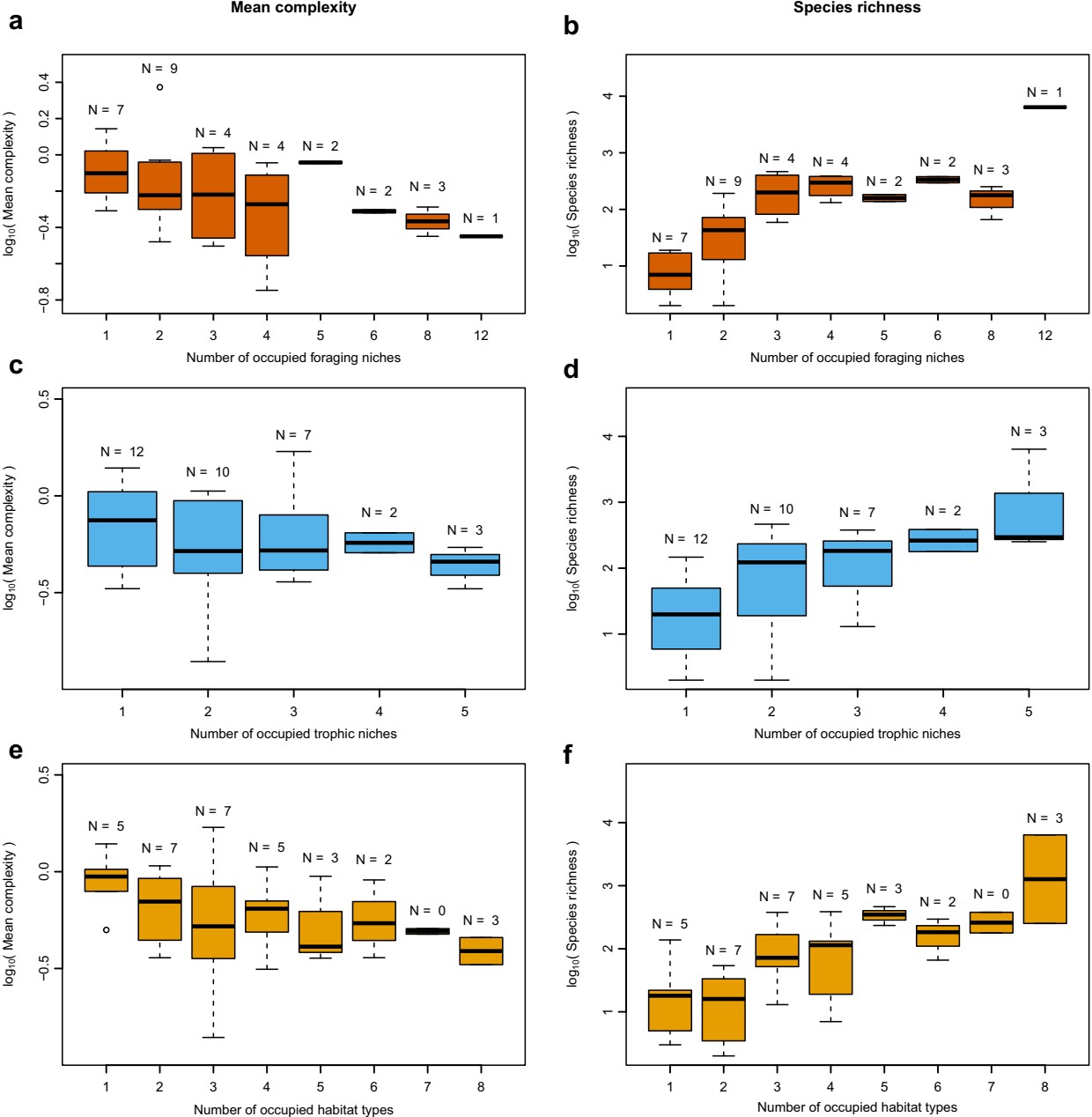

**Fig. 5 | Pairwise comparisons of the number of foraging niches, trophic niches, and habitat types occupied by clades and either their mean complexity or species richness.** Mean complexity score and species richness were $\log_{10}$-transformed. The negative correlations between clade complexity and **a)** the number of foraging niches, **c)** the number of trophic niches, and **e)** the number of habitat types are shown, and the correlations with foraging niche and habitat type are significant and marginally significant, respectively. The positive correlations between clade species richness and **b)** the number of foraging niches, **d)** the number of trophic niches, and **f)** the number of habitat types are shown. All of these comparisons with species richness are statistically significant. The number of clades (N) that occupy each number of categories are indicated in each panel. The boxplots show the median, upper and lower quartile, and upper and lower extreme of mean complexity and species richness for the clades that occupy each number of different categories. Outliers are indicated.

ecology and evolution, we were able to test the correlation of complexity scores with Kipp's distance, finding a weak, but significant, positive correlation (Supplementary Fig. 2). Kipp's distance is a potential predictor of dispersal ability given its impact on aerodynamics and flight. However, it is only a weak predictor, and even then only when modelled in conjunction with other variables[52], for which we did not have data across our sample. Furthermore, it is difficult to disentangle the effects of Kipp's distance from the effects of varying migratory behaviour[52]. Links between morphological

complexity and dispersal ability may yet prove to be of importance, but require extensive further work to verify.

Crouch & Tobias[39] reported a strong positive correlation between the rate of evolutionary change in habitat-type and the species richness of clades in extant birds, as well as an association between ecological and dietary stability and more rapid phenotypic evolution in certain morphological traits. Their findings are congruent with ours, in that we find greater species richness in more ecologically diverse clades with more "conservative" morphologies (i.e., less differentiation

amongst parts, and therefore lower complexity). Similarly, we find lower species richness in more ecologically constrained clades with more "unusual" morphologies (i.e., greater differentiation amongst parts, and therefore greater complexity). Primates with more unique and derived limb skeleton morphologies have been shown to be less evolvable in simulation, and to be associated with higher rates of extinction while evolving between different ecomorphological specialisations[42,43], echoing our similar observations in birds.

Predictions of a positive relationship between complexity and species richness posited by Doebeli & Ispalotov[35] may not be realised here for another, less biological reason. These authors defined complexity in terms of the number of axes of variation, while our study system has a fixed number of variable parts (although it does capture the extent and structuring of differentiation within those parts). This difference highlights the conceptual difficulties inherent in quantifying and studying complexity. Complexity cannot be measured directly, and indices inevitably quantify those aspects of morphological variation that are most amenable to quantification. This can vary significantly across the tree of life, and across anatomical modules in the same organisms.

A structure such as the vertebral column may be more amenable to Doebeli & Ispalotov's[35] framework, as it has a variable number of parts which themselves have variable form. McShea[6] and Fusco & Minelli[62] presented composite indices that can be derived from simple metrics of between-element serial differentiation, and that have been applied elsewhere in studies of complexity in the vertebral column of mammals[6,63]. Information theoretic indices, such as the Brillouin index[64], are commonly used to summarise the total number of parts present in a physical structure, and their coarse differentiation into part types. This approach has been applied to the limb pairs of arthropods[65,66], and the vertebral columns of mammals[67]. Even simple counts of the number of parts and ratios of their relative frequency are useful in some contexts[67,68].

The skeleton of birds is highly modular[15,69,70]. Although some degree of covariance is expected between modules[50], modules typically evolve at different rates and in response to different selective regimes (e.g., refs. 56,70). We do not expect that different regions of anatomy or methods of quantifying complexity should necessarily yield congruent results. Combining data from multiple skeletal elements would allow a more holistic quantification of morphological complexity, especially given the proposed importance of modularity in the evolution of biological complexity[71]. Correlated evolution between the hindlimbs and the cervical morphology of birds has already been documented[72], and so quantifying and studying the complexity of the avian vertebral column offers a logical point of departure for future analyses.

Our results demonstrate evidence of complexity-mediated constraints on the potential for ecological diversification, and by extension on the propensity for clades to diversify taxonomically. While this is congruent with predictions made by Orr[37], we do not explicitly examine evolutionary rates and how they relate to morphological complexity, and so we cannot directly test his predictions. However, our results are consistent with slower rates of adaptation into novel dietary and habitat niches among more complex species and clades, as well as lower diversity. In this context, we note that the most iconic adaptive radiations among extant birds occur in the passerines (e.g., Darwin's finches, Hawaiian honeycreepers, Madagascan vangas[40,41]), which have a relatively low mean limb skeletal complexity (Fig. 3).

Testing the relationship between complexity and net diversification rate will be a critical next step. It will also be important to test whether elevated rates of change in morphological complexity correlate with periods of adaptive diversification.

## Methods
### Obtaining species-level morphological and trait data
Measurements of limb bone lengths were taken from the literature. References are provided in Supplementary Data 1 and Supplementary

References, and institutional abbreviations are given in Supplementary Table 1. A small number of specimens were consulted first hand in the collections of the University Museum of Zoology (UMZC), Cambridge, UK (see Supplementary Data 1), and were measured using Mitutoyo DC-8" AX (8"/200 mm) digital callipers. Where multiple specimens were documented for each species, the mean length of each element for each species was taken. Juvenile specimens were discounted where they were identified as such in their source publication. Sexual dimorphism was not used as a variable, as sex was not always recorded in our sources. When not reported, the length of the ulna was inferred from the length of the radius using phylogenetic Reduced Major Axis regression (pRMA), implemented in the phytools package[73] in R v4.1.2[74]. We have confidence in this, given strong integration in size between the radius and ulna across birds[50], and the high proportion of variance explained by the pRMA model (pRMA, $R^2 = 0.959$, $p = 1.0 \times 10^{-6}$, $\lambda = 0.829$; Supplementary Fig. 6). The model was constructed using data for species that possessed data for both the radius and ulna, which account for approximately 2/3 of the sample ($n = 643$; Supplementary Fig. 6). A total of 161 species out of 983 had ulnar lengths inferred in this way. RMA is preferred to ordinary least squares due to its symmetric nature – we have no belief that ulnar length is determined by radial length, or vice versa. Full sets of measurements were obtained for 983 species, representing 35 extant orders, and are summarised in full in Supplementary Data 1.

Estimates of species body mass in grams were taken from the AVONET database[75], as were Kipp's distances and classifications of trophic niche and habitat type. We obtained mass estimates for all 983 species in our sample, trophic niche classifications and Kipp's distances for 870 species, and habitat type data for 869. More detailed classifications of foraging niche categories were taken from Pigot et al.[55], and were available for 773 species (Supplementary Data 1).

Estimates of species richness for bird orders were taken from Birds of the World online (birdsoftheworld.org[21]; Supplementary Data 2). These were $\log_{10}$-transformed for use in our analyses.

### Phylogeny
We used the species-level phylogeny of Cooney et al.[76] as the basis for all phylogenetically informed analyses. Their tree was produced by transplanting species-level clade topologies taken from Jetz et al.[19,20], and grafting them onto the more strongly supported molecular backbone produced by Prum et al.[77]. This composite tree has the advantage of being sampled comprehensively at the species-level, and having a well-supported backbone of major clade relationships. It was subsequently pruned to match the sample of species for which morphological data were available (Supplementary Fig. 7; Supplementary Code 1).

An ordinal-level tree was also produced for analyses performed at that higher level of taxonomy (Supplementary Code 1). This was achieved by further pruning the tree from Cooney et al.[76] to leave ordinal level clades as the tips (Fig. 3). All processing of trees was performed using functions in the phytools package[73] for R[74].

### A proxy of complexity
To derive a suitable proxy for limb skeleton complexity, we first had to remove all variations in bone lengths that were related to variable species body size. Body mass is a commonly cited variable that correlates with many aspects of organismal biology[78], including influences on abundance at higher taxonomic levels[79], extinction[80,81], taxonomic diversity[82], and rates of evolution of forelimb morphology in early birds[83]. It also exhibits isometric and allometric scaling with certain aspects of avian osteology[50,84,85], which we observed within our sample (Supplementary Fig. 1). Hence, it is critical to control, so that macroevolutionary patterns related to body size do not obscure trends related to complexity. The convention for achieving this is to first log-transform the raw values to achieve a data distribution appropriate for

parametric statistical procedures (see Benson et al.[86]), and then perform a phylogenetic regression (PGLS[87,88]) of these values against body mass estimates. The residuals from this model can then be used for further analyses.

However, we decided not to follow this procedure here. Log-transformation of the raw values affects their scaling in a non-linear fashion, such that numeric values representing bone lengths for larger animals would be transformed by a different proportion. This is not desirable, as complexity is defined in part as depending on relative differentiation of parts within a given species, such that isometrically scaled versions of the same configuration of parts and relative differentiation should yield the same complexity score (Supplementary Note 1). To avoid this, we transformed the raw measurements to remove isometric size variation using a different method. We did, however, repeat all analyses using the conventional approach, and found all results to be consistent across both approaches (Supplementary Tables 5–8).

Firstly, the mean length of all 6 measured elements (femur, tibiotarsus, tarsometatarsus, humerus, ulna, and carpometacarpus) for each species was calculated, and the length of each element was expressed as a proportion of that mean. Therefore, if all bones were of equal length, all would yield a scaled value equal to one. This has the effect of entirely removing any notion of the absolute size of the animal, while retaining information regarding the relative length differences for each bone, and each limb pair. Comparison to an alternative method of scaling, whereby lengths were transformed to represent a proportion of the mean length of their respective limb pair, showed this retention of information to be true, as they resulted in different complexity scores (Supplementary Note 1; Supplementary Fig. 8). We opted for an isometric transformation approach, as log-transformation entailed undesirable properties (Supplementary Note 1; Supplementary Table 9). However, this scaling takes no account of allometric variation. To do this, we input the isometrically-transformed limb bone lengths into a phylogenetic regression against estimates of species body mass, and took the phylogenetic residuals from that model for use in further steps. The model was implemented using the phyl.resid function in the phytools package[73] for R. Body mass estimates were $\log_{10}$-transformed for use in this model.

To derive a univariate index of complexity from these phylogenetic residual relative mean bone lengths, we first plotted them in a three-dimensional space. Each axis represented variation in a different equivalent segment of the limb pairs – either the autopod, stylopod, or zeugopod. In this space, each species was represented by two points, one representing each limb pair. Coordinates on each axis were determined by the values of the phylogenetic residuals derived in the procedure described above. The complexity of each species was taken as the Euclidean distance between these two points. In the hypothetical least complex possible configuration of osteological elements – one where both limb pairs are of equal length with no differences in the proportions of equivalent elements – both points for that species would plot directly on top of one another, yielding a complexity index of zero. Any differentiation in the relative lengths of the limb pairs, or the relative lengths of the equivalent elements between each pair, will cause the points to diverge in the space and increase the index of complexity. Manhattan distances were calculated in addition to Euclidean distances. The two indices correlated closely (PGLS, $p = 2.2 \times 10^{-16}$, F = $2.285 \times 10^4$ on 1 & 981 d.f., adj.$R^2$ = 0.9588, $n$ = 983; Supplementary Fig. 9) and yielded congruent findings (Supplementary Tables 5-8). The precise index used (Euclidean or Manhattan) therefore had minimal impact on our results or their interpretation.

These data offer indices of complexity in the "horizontal" sense, comparable between entities (in this case species) at the same hierarchical level of their structural organisation[5,12].

## Phylogenetic modelling of clade complexity, species richness, and ecology

The mean complexity score was calculated for each ordinal-level clade. These mean scores required $\log_{10}$-transformation to achieve a normal distribution of data that was appropriate for use in PGLS analysis. PGLS analysis of clade species richness and mean complexity were performed using functions in the caper package[89] for R. The ordinal-level tree described above was used to generate the phylogenetic variance-covariance (VCV) matrix. Phylogenetic signal was estimated using the K statistic[51] under maximum likelihood as part of the model fitting procedure. The Opisthocomiformes were removed from this analysis, as they are represented by only a single extant species (*Opisthocomus hoazin*).

PGLS analyses were also performed at the species-level to investigate correlations between the complexity scores for species and their occupied foraging, trophic, and habitat niches. These were fitted separately, comparing complexity scores for species against the categorical data describing their ecological variation. The models were fitted using functions in the caper package, and phylogenetic signal was again estimated using the K statistic[51]. The species-level phylogeny described above was used for generating the phylogenetic VCV matrix. *Opisthocomus hoazin* was retained in these species-level PGLS models. However, the single desert-dwelling species (*Chlamydotis undulata*, Houbara Bustard) was removed from the habitat model, to avoid single-species groups and sample-size artefacts that they engender. No other habitat type categories, or any trophic or foraging niche category, contained single species, so no further pruning was necessary.

Further PGLS analysis was performed on $\log_{10}$-transformed complexity scores and the square-root of Kipp's distances. The square root was favoured, as it yielded a distribution of scores that best conformed to the assumptions of the test. A total of 9 species were pruned as outliers, as they compromised the reliability of the model fit. These were identified as points lying beyond 3 standard deviations of the mean for either variable. This model was again fitted using functions in the caper package, with phylogenetic signal estimated under maximum likelihood with the K statistic.

## Analysis of the extent of ecological variation in relation to clade species richness and mean complexity

For each ordinal-level clade, the number of different foraging niche, trophic niche, and habitat type categories occupied by the sampled constituent species were counted. These counts were used as explanatory variables for either mean clade complexity (again, $\log_{10}$-transformed prior to analysis) or clade species richness in pairwise PGLS analyses, performed using functions in the caper package for R. *Opisthocomus hoazin*, as the only representative of Opisthocomiformes, was removed to avoid single-species groups.

We used a randomisation approach[90,91] to provide a null expectation for the relationship between ecological specialisation and clade species richness. We might expect that more diverse clades will occupy a greater number of niches by virtue of their greater sample size. By the same token, the most species-poor clades cannot achieve the maximum possible levels of ecological diversity, since they contain fewer species than there are ecological categories in the data. We began by randomly reassigning tip values of each of the three categorical variables (foraging niche, trophic niche, and habitat type) into a number of groups equal to those in the empirical analyses, and of the same sizes as used in the empirical analyses. This was repeated 1000 times for each variable. The PGLS approach described above was repeated across all 1000 randomised replicates of each variable, and the $p$-values, y-intercept and slope coefficients, and adjusted $R^2$ values of the models were collated to build null distributions of the correlations expected by chance. A $p$-value can be assigned to these by measuring the proportion of randomly-obtained values which are more extreme than the value obtained empirically. If the empirical value falls within

or beyond the 5% most extreme random values, we infer a significant result, as the empirical finding deviates substantially from the null distribution. It is important to note that this is not an exact $p$-value, and is limited by the number of random data permutations which are performed. In this case using 1000 permutations, the minimum possible $p$-value is 0.001, or $p < 0.001$ if the empirical observation is more extreme than all random ones.

Further PGLS models, in which both mean complexity and number of occupied ecological categories were used as predictors of clade species richness in an additive manner, were also fitted. The desert-dwelling category was removed from the model containing habitat type as a variable, as it was sampled by only a single species.

In all cases the ordinal-level phylogeny was used to generate the phylogenetic VCV matrix, and K was estimated under maximum likelihood as a measure of phylogenetic signal in the data. This was implemented using functions in the caper package for R.

### Reporting summary
Further information on research design is available in the Nature Portfolio Reporting Summary linked to this article.

## Data availability
The morphological, ecological, phylogenetic, and species richness data used in this study have been deposited in Figshare (10.6084/m9.figshare.23941488). All data necessary for replication of the analyses and figures in this work are provided in the files available from this repository.

## Code availability
All data manipulation and analyses were conducted using custom R scripts which draw upon other existing R software packages, as referenced in Methods. These can be used to replicate all analyses and figure plotting, and are provided in Figshare: 10.6084/m9.figshare.23941488.

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

## Acknowledgements
This work was supported by funding from the John Templeton Foundation (Grant 61408) to MAW and MR, and from a NERC GW4 + DTP Studentship 2481350 to AB and MAW. MAW's work was also supported by BBSRC grants BB/K015702/1 and BB/K006754/1. We thank Roger Benson for enormously helpful feedback, enabling us to substantially improve this paper. We also thank Daniel J. Field and Matthew Lowe for granting access to collections at the University Museum of Zoology (UMZC, Cambridge, UK).

## Author contributions
A.B. and M.A.W. conceived the study. All authors (A.B., E.G., Y.L., J.O., M.R., M.A.W.) contributed to the experimental design. A.B. collected and analysed the data. A.B. wrote the initial draft of the manuscript, to which all other authors (E.G., Y.L., J.O., M.R., M.A.W.) provided comments and revisions.

## Competing interests
The authors declare no competing interests.
