## [Peer Review File · Nature Communications]

REVIEWER COMMENTS

Reviewer #1 (Remarks to the Author):

In this paper the authors describe analyses of the link between morphological complexity and diversity based on a substantial and broad sample across the extant avian tree of life. The results suggest that there is a link between complexity and 1) the number of dietary and habitat niches and 2) species richness. In particular, species richness declines with increasing complexity.

I enjoyed reading the manuscript and in particular the excellent scholarship around the core concepts and predictions linking complexity to diversity. I have been introduced to an interesting range of literature as a result. I found the premise of the paper fascinating and likely to be of broad interest. However, I am not convinced that the data and metric used as a proxy for complexity are adequate. Broadly, complexity describes the number of parts and differentiation of those parts. The metric here (in simple terms) describes relative proportions and differences among limb length pairs. The lowest complexity score arises when all of the six measured bones are the same length. But, biologically, it is not at all clear that this is relevant. From a developmental or genetic perspective it is difficult to understand why equal bone lengths represent more complex processes than bones of unequal length. Similarly, there is no reason to assume from an ecological perspective that equal bone lengths are more complex than unequal lengths. The limb length proportion metric may well capture something interesting biologically - indeed the phylogenetic distribution of the complexity score suggests that there may be some ecological relevance. The problem is that there is a lack of justification for referring to this as complexity. A case could be made that complexity is partially related to dispersal. The most complex taxa seem to be large, long legged groups that may be poor dispersers or longer winged species that are wide dispersers. In this sense complexity may reflect extremes of dispersal ability with low complexity associated with intermediate dispersal. It may then be that the illustrated link between the metric of complexity and species richness is revealing something about correlation between intermediate dispersal ability and speciation.

Measuring complexity is clearly challenging and directly addressing whether a given metric is a good proxy for complexity may not be possible. However, I do think it is important to 1) more strongly justify why this is a relevant metric to use for measuring complexity and 2) examine whether the metric is associated with other traits in more depth. As highlighted above, one such trait is dispersal but it would also be helpful to understand broader links to ecology and life history. Without this deeper understanding of what the complexity score measures, it is not possible to adequately interpret what is potentially a very interesting relationship.

Minor comments

Line 421: phylogenetic signal was estimated with K. I assume that this is Blomberg's K, if so the paper should be cited (<https://onlinelibrary.wiley.com/doi/10.1111/j.0014-3820.2003.tb00285.x>). A picky point, K is a statistic rather than a parameter.

Reviewer #2 (Remarks to the Author):

Paper Review Nature Comms 391737

This paper quantifies the relationship between complexity and diversity across extant bird clades with an impressive dataset of nearly 1000 species, demonstrating a negative correlation between morphological complexity and taxonomic diversity. This is an interesting finding and grows our understanding of the factors that influence differential diversity across the tree of life. With only minor revisions this will be an excellent contribution to Nature Communications.

The authors claim to test two alternative hypotheses, first that complexity is positively correlated with richness because it promotes more rapid diversification and second that complexity is negatively correlated with diversity because it confers greater risk of extinction. Because they are

testing the correlation between complexity and richness and not complexity~speciation rate nor complexity~extinction risk, I think it would be better to reduce the text hypothesizing mechanisms driving either pattern until the discussion section. Indeed, in the discussion you hypothesize that the negative complexity/richness pattern is explained by "complex species should be slower to evolve and adapt" rather than extinction risk

It is clear from Figure 4 that Passeriformes is a major outlier for log(species richness). I used the supplemental code to re-run the analyses with this clade removed and I was happy to see that the overall relationship between diversity and complexity was unchanged. I think it would be good to acknowledge this outlier and that it is not strongly biasing the results in the main manuscript text.

On line 89 you say: "Considered together, these variables allow us to index both the "differentiation" and "arrangement" components of morphological complexity (sensu [5-8]), with the number of constituent parts remaining constant. " After this part of the manuscript you never again return to arrangement component, instead focusing only on arrangement. I think it would be best to focus only on differentiation in the introduction.

Line 66-67: "including Darwin's finches and Hawaiian honeycreepers, and Madagascan vangas" should be "including Darwin's finches, Hawaiian honeycreepers, and Madagascan vangas [40,41].". the comma is necessary as the finches are Darwin's but the honeycreepers are not.

Line 89: I think it would be helpful to define "differentiation" and "arrangement" in the present paper

There seems to be a small problem with the supplemental R code file. On lines 119 and 120 I get the following error message:

```
Error in which(rownames(logged_lengths_phylmod$resid) == names(logged_comp_scores)[i]) :  
object 'logged_comp_scores' not found
```

It looks like you are calling an object named "logged_comp_scores" that was never defined. Double check before submitting.

RESPONSE TO REVIEWERS' COMMENTS

Reviewer 1

- *"In this paper the authors describe analyses of the link between morphological complexity and diversity based on a substantial and broad sample across the extant avian tree of life. The results suggest that there is a link between complexity and 1) the number of dietary and habitat niches and 2) species richness. In particular, species richness declines with increasing complexity."*

Yes, this is a fair and accurate statement of the scope of our study, and of our key findings.

- *"I enjoyed reading the manuscript and in particular the excellent scholarship around the core concepts and predictions linking complexity to diversity. I have been introduced to an interesting range of literature as a result. I found the premise of the paper fascinating and likely to be of broad interest."*

We sincerely thank the referee for their kind words.

- *"However, I am not convinced that the data and metric used as a proxy for complexity are adequate. Broadly, complexity describes the number of parts and differentiation of those parts. The metric here (in simple terms) describes relative proportions and differences among limb pairs. The lowest complexity score arises when all of the six measured bones are the same length. But, biologically, it is not at all clear that this is relevant."*

The referee highlights some ambiguity in our writing – we have recast this for much greater clarity. The referee is also correct, that the score is at a minimum if *"all of the six measured bones are the same length"*. However, the identity of all six bones is not what is being measured here, but rather the difference between fore and hindlimbs (which will be zero if all bones are identical). Importantly, the minimum score can also be achieved when there is heterogeneity *within* limbs, but both limb pairs are of exactly equal length and proportions. We therefore treat fore- and hindlimbs as serial homologues, and are interested in the extent to which they are differentiated. Differences between the lengths of bones within a limb do not contribute directly to the indices, other than to the extent that these contrast with the lengths of the elements in the other limb pair. Differentiation between limb pairs has well established biological implications (we detail this in the manuscript, with references cited).

- *"From a developmental or genetic perspective it is difficult to understand why equal bone lengths represent more complex processes than bones of unequal length."*

As above, our original explanation was not clear enough, and we have rewritten the relevant sections. Our null assumption is for homogeneity wherever elements (limb pairs, bones, genes, etc) arise by duplication or are otherwise homologous. Our serial homologues are the limbs, not the bones *within* a limb. Such homogeneity characterises low (rather than high) complexity. The zero-force evolutionary law (McShea & Brandon, 2010) conceptualises complexity in this way. Developmental differences underpin the differentiation that leads to greater complexity, and these differences are known and understood in birds. We have expanded the text, and cited a recent review of limb development which includes avian case studies (line 114).

- *"Similarly, there is no reason to assume from an ecological perspective that equal bone lengths are more complex than unequal length. The limb length proportion metric may well capture something interesting biologically - indeed the phylogenetic distribution of the*

complexity score suggests that there may be some ecological relevance. The problem is that there is a lack of justification for referring to this as complexity.”

We should have discussed possible relationships between morphological complexity (which all our indices seek to capture) and functional complexity in our manuscript. There is no expectation that morphological and functional complexity should be correlated. We have added a section justifying our distinction between morphological complexity and functional complexity at lines 34-39. Despite this theoretical separation, we do recover a significant relationship between complexity and various proxies for ecology, implying that our metric does capture morphological variation in an ecologically relevant manner. We cite several references in the manuscript which show evidence of the impact of limb lengths and proportions on various aspects of ecology and life history.

- *“A case could be made that complexity is partially related to dispersal. The most complex taxa seem to be large, long legged groups that may be poor dispersers or longer winged species that are wide dispersers. In this sense complexity may reflect extremes of dispersal ability with low complexity associated with intermediate dispersal. It may then be that the illustrated link between the metric of complexity and species richness is revealing something about correlation between intermediate dispersal ability and speciation.”*

This is entirely possible, although not something we set out to address in our original manuscript. While we do not have data on dispersal ability *per se*, we were able to collect data on Kipp’s distance (the distance between the tip of the first secondary flight feather and the tip of the longest primary flight feather) as a first approximation. This has been used in models elsewhere, in conjunction with other variables, as a proxy for dispersal ability (Dawideit *et al.*, 2009). We have added an additional PGLS analysis in the Results (lines 192-199) and Materials & Methods (lines 482-487) sections (including a new Supplementary Figure), and a more explicit examination of this possibility in the Discussion section at lines 309-317. This expands on the original text. We stress that model fit is very poor, likely reflecting the complexity of the relationship and the influence of many other variables for which we do not have data. A great deal of further work, well beyond the scope of this study, would be needed to unpack it.

- *“Measuring complexity is clearly challenging and directly addressing whether a given metric is a good proxy for complexity may not be possible. However, I do think it is important to 1) more strongly justify why this is a relevant metric to use for measuring complexity...”*

See above points, and additions/alterations to the manuscript at lines 34-39 and 114.

- *“... and 2) examine whether the metric is associated with other traits in more depth. As highlighted above, one such trait is dispersal but it would also be helpful to understand broader links to ecology and life history. Without this deeper understanding of what the complexity score measures, it is not possible to adequately interpret what is potentially a very interesting relationship.”*

This would entail an enormous amount of additional data collection and modelling, as few such traits are consistently quantified at this level of sampling. This would effectively entail writing an entirely new paper. As detailed above, we have added an additional analysis and Supplementary Figure measuring the correlation between complexity and Kipp’s distance (a weak proxy for dispersal ability). What the reviewer suggests is certainly a good prospect for future investigations.

- *“Line 421: phylogenetic signal was estimated with K. I assume that this is Blomberg’s K, if so the paper should be cited (<https://onlinelibrary.wiley.com/doi/10.1111/j.0014-3820.2003.tb00285.x>). A picky point, K is a statistic rather than a parameter.”*

A citation of Blomberg *et al.*’s paper has been added where a K statistic is first reported on lines 161-162. References to K as a parameter have been altered to refer to it as a statistic (these on lines 468 and 475).

Reviewer 2

- *“This paper quantifies the relationship between complexity and diversity across extant bird clades with an impressive dataset of nearly 1000 species, demonstrating a negative correlation between morphological complexity and taxonomic diversity. This is an interesting finding and grows our understanding of the factors that influence differential diversity across the tree of life. With only minor revisions this will be an excellent contribution to Nature Communications.”*

This is an accurate statement of our central goal and finding, and we thank the reviewer for their kind words.

- *“The authors claim to test two alternative hypotheses, first that complexity is positively correlated with richness because it promotes more rapid diversification and second that complexity is negatively correlated with diversity because it confers greater risk of extinction.*

The second alternative is related to a theoretical work by Orr (2000). That work pertains only to the rate of adaptive evolution, and as such could relate to either speciation or extinction rates. We state that this could impact diversity in more complex clades by either slowing the rate of diversification, or increasing the risk of extinction, or both. Whether one of these mechanisms is prevalent over the other in this sample is not currently clear, and so in the Discussion we do not attempt to make direct links to either process (see next comment). We instead explain our findings in the context of their congruence to Orr’s original models, which assess only rates of adaptive evolution.

- *“Because they are testing the correlation between complexity and richness and not complexity~speciation rate nor complexity~extinction risk, I think it would be better to reduce the text hypothesizing mechanisms driving either pattern until the discussion section. Indeed, in the discussion you hypothesize that the negative complexity/richness pattern is explained by “complex species should be slower to evolve and adapt” rather than extinction risk”*

This is a fair point, but we hesitate to remove it from the Introduction, as we consider it to be key in formulating and justifying the alternative hypotheses we propose. The reviewer is correct in stating that we do not directly assess the relationship between either speciation or extinction rates, which we acknowledge in the Discussion as a major opportunity for future work.

- *“It is clear from Figure 4 that Passeriformes is a major outlier for log(species richness). I used the supplemental code to re-run the analyses with this clade removed and I was happy to see that the overall relationship between diversity and complexity was unchanged. I think it would be good to acknowledge this outlier and that it is not strongly biasing the results in the main manuscript text.”*

This is an excellent point. We have changed the Results section at lines 165-169 to acknowledge Passeriformes as an outlier, and provide summary statistics obtained from repetition of model-

fitting with passerines excluded. Further summary statistics and parameters from this refitted model have been appended to SI Table 4 in the Supplementary Information document.

- “On line 89 you say: ‘Considered together, these variables allow us to index both the ‘differentiation’ and ‘arrangement’ components of morphological complexity (sensu [5-8]), with the number of constituent parts remaining constant.’ After this part of the manuscript you never again return to arrangement component. I think it would be best to focus only on differentiation in the introduction.”

We have removed sections of the text referring exclusively to “arrangement” from the Introduction. We concur that this makes for a clearer and more streamlined paper.

- “Line 66-67: “including Darwin’s finches and Hawaiian honeycreepers, and Madagascan vangas” should be “including Darwin’s finches, Hawaiian honeycreepers, and Madagascan vangas.”

Agreed. This alteration has been made at line 72.

- “Line 89: I think it would be helpful to define “differentiation” and “arrangement” in the present paper.”

An excellent point. We have added text at lines 99-101. “Differentiation” as applied to serially homologous structures within an individual organism is analogous to the disparity of species within a clade, and could be quantified using the same indices as deployed for the latter purpose. As “arrangement” is no longer discussed we do not provide a working definition for it.

- “There seems to be a small problem with the supplemental R code file. On lines 119 and 120 I get the following error message:

```
Error in which(rownames(logged_lengths_phylmod$resid) ==  
names(logged_comp_scores)[i]) : object 'logged_comp_scores' not found
```

It looks like you are calling an object named “logged_comp_scores” that was never defined. Double check before submitting.”

This reported bug has been checked and fixed. The object was defined as “logged_scores”, then called as “logged_comp_scores”. It is now consistently referred to as “logged_comp_scores”.

References

- Dawideit, B. A., Phillimore, A. B., Laube, I., Leisler, B., & Böhning-Gaese, K. (2009). Ecomorphological predictors of natal dispersal distances in birds. *Journal of Animal Ecology*, 78(2), 388-395. doi:<https://doi.org/10.1111/j.1365-2656.2008.01504.x>
- McShea, D. W., & Brandon, R. N. (2010). *Biology's First Law*. Chicago & London: The University of Chicago Press.
- Orr, H. A. (2000). Adaptation and the cost of complexity. *Evolution*, 54(1), 13-20. doi:<https://doi.org/10.1111/j.0014-3820.2000.tb00002.x>

REVIEWER COMMENTS

Reviewer #1 (Remarks to the Author):

In the revised manuscript and response letter the authors have clarified a number of points around the complexity measure and its meaning, correcting misunderstandings on my part. I am now far more convinced by their choice of metrics. Given this greater clarity, my other suggestion around the association of the complexity with other metrics is much less important and I agree with the authors that it is not necessary for this manuscript. I have a small number of minor suggestions, including one additional (but straightforward) analysis but otherwise .

1. Lines 49-50: the authors suggest that most comparisons of diversity and disparity have been implemented in a palaeo context. I agree that this is largely true in a macroevolutionary context but in ecology there are many studies in a spatial context of functional diversity and species richness. Functional diversity and disparity are the same metric (or toolbox of metrics) but given different names in different disciplines. This is a longwinded way of suggesting being explicit about the macroevolutionary context in this sentence.

2. Line74-75: The authors suggest that Orr's prediction is more intuitively linked to extinction than speciation. This is subjective and it could be argued that the link to speciation is intuitive in the context of Schluter's idea of evolution along line of least resistance.

3. Line 94: extra word - remove "but"

4. Section beginning line 215: there are strong positive correlations between measures of habitat or niche diversity and species richness. This seems inevitable by chance alone (small clades cannot attain the highest levels of niche diversity). The null expectation is not of no relationship. A rough null expectation could be derived by random shuffling of niche and habitat among tips (species) and then recalculating clade niche and habitat diversity. This randomisation can be repeated and used to establish a null expectation for relationship between niche number and species richness. This would give context to the observed results.

Reviewer #2 (Remarks to the Author):

The authors have adequately addressed the concerns raised by the reviewers in the previous round of reviews. I will be happy to see this accepted and published.

RESPONSE TO REVIEWERS' COMMENTS

Reviewer 1

"In the revised manuscript and response letter the authors have clarified a number of points around the complexity measure and its meaning, correcting misunderstandings on my part. I am now far more convinced by their choice of metrics. Given this greater clarity, my other suggestion around the association of the complexity with other metrics is much less important and I agree with the authors that it is not necessary for this manuscript. I have a small number of minor suggestions, including one additional (but straightforward) analysis but otherwise ."

- We are glad that we were able to improve the clarity of the manuscript, and thank the reviewer for bringing these points to our attention. This has undoubtedly improved the quality of the work greatly.

"1. Lines 49-50: the authors suggest that most comparisons of diversity and disparity have been implemented in a palaeo context. I agree that this is largely true in a macroevolutionary context but in ecology there are many studies in a spatial context of functional diversity and species richness. Functional diversity and disparity are the same metric (or toolbox of metrics) but given different names in different disciplines. This is a longwinded way of suggesting being explicit about the macroevolutionary context in this sentence."

- We have added to the relevant text (lines 49-50) to make clear that we are referring solely to studies in a macroevolutionary context.

"2. Line 74-75: The authors suggest that Orr's prediction is more intuitively linked to extinction than speciation. This is subjective and it could be argued that the link to speciation is intuitive in the context of Schluter's idea of evolution along line of least resistance."

- This point is fairly made, and we have removed the subjective language. This has been replaced by a more direct statement, avoiding any notion of intuition (line 76).

"3. Line 94: extra word - remove "but""

- The extra word has been removed.

"4. Section beginning line 215: there are strong positive correlations between measures of habitat or niche diversity and species richness. This seems inevitable by chance alone (small clades cannot attain the highest levels of niche diversity). The null expectation is not of no relationship. A rough null expectation could be derived by random shuffling of niche and habitat among tips (species) and then recalculating clade niche and habitat diversity. This randomisation can be repeated and used to establish a null expectation for relationship between niche number and species richness. This would give context to the observed results."

- We have done as the reviewer suggested, and implemented a randomisation-like approach, whereby tip values for each of the three categorical variables are randomly reassigned into groups of the same number and sizes as the empirical observations. Refitting of the PGLS models across 1,000 permutations of each variable revealed that significant positive correlations are indeed the null expectation, as the reviewer suspected. We agree that this does provide useful context for this section of the Results. Additional text has been added in the Results (lines 226-264), Discussion (lines 321-322), and Materials & Methods (lines 521-538) sections, as well as three new supplementary figures (SI Figs. 5-7).

Reviewer 2

“The authors have adequately addressed the concerns raised by the reviewers in the previous round of reviews. I will be happy to see this accepted and published.”

- No new points need addressing from this reviewer’s response. We are glad that we were able to adequately respond to and solve their previous concerns, and thank them for the improvements to this work which their comments elicited.